

# Integrating palaeoclimate time series with rich metadata for uncertainty modelling: strategy and documentation of the PalMod 130k marine palaeoclimate data synthesis

**Lukas Jonkers[2], Olivier Cartapanis[2,a], Michael Langner[1], Nick McKay[3], Stefan Mulitza[1], Anne Strack[1], and Michal Kucera[1]**

[1]MARUM – Center for Marine Environmental Sciences, University of Bremen, Bremen, Germany
[2]Institute of Geological Sciences and Oeschger Centre for Climate Change Research, University of Bern, Bern, Switzerland
[3]School of Earth and Sustainability, Northern Arizona University, Flagstaff, Arizona, USA
[a]now at: Aix-Marseille Université, CNRS, IRD, Collège de France, INRAE, CEREGE, Europôle de l'Arbois, Aix-en-Provence, France CE1

**Correspondence:** Lukas Jonkers (ljonkers@marum.de)

**Abstract.** TS1 Palaeoclimate data hold the unique promise of providing a long-term perspective on climate change and as such can serve as an important benchmark for climate models. However, palaeoclimate data have generally been archived with insufficient standardisation and metadata to allow for transparent and consistent uncertainty assessment in an automated way. Thanks to improved computation capacity, transient palaeoclimate simulations are now possible, calling for data products containing multi-parameter time series rather than information on a single parameter for a single time slice. Efforts are underway to simulate a complete glacial–interglacial cycle using general circulation models (https://www.palmod.de/ TS2), and to confront these simulations with palaeoclimate data, we have compiled a multi-parameter marine palaeoclimate data synthesis that contains time series spanning 0 to 130 000 years ago. We present the first version of the data product that focuses exclusively on time series for which a robust chronology based on benthic foraminifera $\delta^{18}$O and radiocarbon dating is available. The product contains 896 time series of eight palaeoclimate parameters from 143 individual sites, each associated with rich metadata, age–depth model ensembles, and information to refine and update the chronologies. This version contains 205 time series of benthic foraminifera $\delta^{18}$O; 169 of benthic foraminifera $\delta^{13}$C; 131 of seawater temperature; 174 and 119 of planktonic foraminifera $\delta^{18}$O and $\delta^{13}$C; and 44, 38 and 16 of carbonate, organic carbon and biogenic silica content, respectively. The data product is available in three formats (R, LiPD and netCDF) facilitating use across different software and operating systems and can be downloaded at https://doi.org/10.1594/PANGAEA.908831 (Jonkers et al., 2019). This data descriptor presents our data synthesis strategy and describes the contents and format of the data product in detail. It ends with a set of recommendations for data archiving.

# 1 Introduction

Global climate has varied dramatically over the last glacial–interglacial cycle. Since the previous interglacial (approximately 130 000 years ago) the Earth had slowly been cooling until the Last Glacial Maximum (LGM; approximately 21 000 years ago). This cooling was associated with the growth of massive ice sheets in North America and Eurasia, leading to a sea level drop of about 120 m (Waelbroeck et al., 2002) and pronounced climate variability on millennial timescales (Voelker and workshop participants, 2002). From the LGM, the Earth warmed rapidly until the onset of the current relatively stable warm period, the Holocene (Shakun et al., 2012). The ultimate cause of the large-scale variations in the Earth's climate is changes in the orbit of the Earth around the Sun (Hays et al., 1976). However, complex feedback and non-linear mechanisms, involving ocean (atmosphere, cryosphere) circulation and biogeochemical cycles, are required to explain how slow changes in the orbital configuration led to the observed evolution of global climate and how these processes led to the manifestation of abrupt climate change.

For these reasons the last glacial–interglacial cycle has been a key target for palaeoclimate modelling. Initially this only involved equilibrium simulations for key time slices, such as the LGM, or transient simulations for short periods, such as the last millennium. The motivation to simulate past climate states is given by the possibility of palaeoclimate data serving as a benchmark for the models. Indeed, this possibility contributed to the development of large palaeo-data syntheses (CLIMAP project members, 1981; MARGO project, 2009). The time-slice modelling approach is still being pursued; for example in phase 4 of the Paleoclimate Modelling Intercomparison Project (PMIP), four of the five target intervals fall into the time frame of the last glacial cycle (Kageyama et al., 2018). However, with increasing computing power, the focus is now shifting towards transient climate simulations (Liu et al., 2009; Latif et al., 2016), and the simulation of the last deglaciation is now also considered in the PMIP protocol (Ivanovic et al., 2016).

This development calls for a different type of palaeodata synthesis, with its focus on time series rather than on time slices. Time series of climate data are needed to evaluate aspects of transient simulations that are not available in equilibrium simulations, such as rates of change, phase relationships and spectral properties of climate variability. It is also clear that an evaluation in multi-parameter space using different aspects of the climate system and multiple proxies will be more powerful and diagnostic (Kurahashi-Nakamura et al., 2017), calling for multi-parameter synthesis products.

Observations of the evolution of past climate are based on proxies (measurable approximations of climate-related variables) and hence are, by definition, indirect. Comparison of proxy-based reconstructions with climate model simulations is therefore far from straightforward, as discrepancies may arise from both model and proxy uncertainty. Proxy uncertainty derives from reconstruction uncertainty (related to calibration, recording bias, archive specifics and instrumental approach) and chronological uncertainty. The latter is particularly relevant to the comparison of transient climate change, and chronological uncertainty thus requires a comprehensive treatment in data syntheses of palaeoclimate time series.

Accounting for proxy uncertainties in a comprehensive and transparent manner requires not only expert knowledge but also the availability of extensive metadata in addition to the proxy data. However, due to a lack of standardisation and inconsistent archiving of metadata, synthesising palaeoclimate data in a way that allows for robust uncertainty assessment remains challenging and time consuming. Efforts are underway to alleviate these challenges. The largest palaeoclimate data repositories (World Data Service for Paleoclimatology, operated by the national centres for environmental information (NCEIs) at NOAA, and PANGAEA) are both striving for more standardisation and to store data in (more) machine-readable formats. In addition, standardisation is progressing through the use of existing data formats from other communities (netCDF; Langner and Mulitza, 2019) as well as the implementation of new data formats specifically targeted to palaeodata (Linked Paleo Data (LiPD); McKay and Emile-Geay, 2016). At the same time there is ongoing discussion on data and metadata requirements and standards (Khider et al., 2019). Traceability of datasets is also improved through data citations, not only ensuring that data producers receive proper credit for their work but also allowing for better linking of different datasets. Nevertheless, these initiatives have only recently been emerging and the majority of the palaeoclimate data remains inconsistently formatted, non-standardised and scattered over various data repositories. The need for synthesis products and documentation of potential synthesis approaches is therefore as large as ever.

Here we present the first version of a new multi-proxy marine palaeoclimate data synthesis that covers the past 130 000 years developed within the German climate modelling initiative PalMod (Latif et al., 2016). We focus on the ocean as it is a large reservoir of heat and $CO_2$ and allows for global coverage with consistent chronological control. This synthesis goes beyond the time frame of many existing multi-proxy and parameter data syntheses (PAGES2k Consortium et al., 2017; Routson et al., 2019), expands existing data products that provide long palaeoclimate time series to multiple parameters (Shakun et al., 2012; Marcott et al., 2013; Peterson and Lisiecki, 2018; Snyder, 2016), and is based on a strategy of semi-automated data harvesting (Cartapanis et al., 2016). This version of the synthesis contains data on nine climate-sensitive parameters: benthic and planktonic foraminifera stable oxygen and carbon isotopes, seawater temperature, radiocarbon and bulk sediment carbonate, organic carbon, and biogenic silica content.

In this paper we describe our synthesis approach, the contents and structure of version 1.0.0 of the data product, suggestions for the product's use, plans for future updates, and recommendations for archiving new data and retrieving dark
[5] data in a way that allows for optimal future reuse. The data product is intended to be used to investigate spatio-temporal changes in a multi-parameter domain. Thanks to rich metadata that allow for us to rigorously quantify reconstruction uncertainties, we also envision that this data product will
[10] provide the building blocks for intelligent palaeoclimate data model comparison (Weitzel et al., 2019), for instance through proxy system modelling (Dolman and Laepple, 2018) or data assimilation (Breitkreuz et al., 2019).

The structure of this data descriptor is as follows. Section 2
[15] describes the synthesis strategy, including the data discovery approach, standardisation and age modelling. In Sect. 3 we provide general information on palaeoclimate proxies from marine-sediment archives that is CE2 used to guide the metadata selection. Section 4 details the structure of the database,
[20] and the contents of version 1.0.0 are outlined in Sect. 5. The formats of the data product and where it can be accessed are described in Sect. 6, and we discuss future plans, versioning and intended use in Sect. 7. In the last section, Sect. 8, we reflect on the data synthesis effort and provide recommenda-
[25] tions for data archiving and data rescue.

## 2 Data synthesis strategy

Our data product focuses on time series from marine-sediment archives. A single marine-sediment archive (sediment core) can be used for measurements of different pa-
[30] rameters, each providing information on different aspects of the environmental conditions at the time of deposition. However, for the purpose of analysis, the various proxy time series must refer to a single age–depth model for the sediment core they are derived from. For this reason, the basis of our
[35] synthesis is formed by a collection of sediment cores, each associated with its own age–depth model.

Marine sediments are dated using absolute age controls, where specific layers are dated using, for instance, radiocarbon, tephra or palaeomagnetic properties, and/or relative age
[40] controls, where time series are aligned based on the hypothesised synchronicity of the changes recorded by some properties of the sediment. A well-established hypothesis-based age modelling approach with a solid theoretical basis is the alignment of benthic foraminifera stable-oxygen-isotope ra-
[45] tio ($\delta^{18}$O) time series (Lisiecki and Raymo, 2005). We thus base our chronological framework on a combination of radiocarbon dates and benthic foraminifera $\delta^{18}$O and have selected time series where both parameters are available as the foundation of this data product. This approach of blending
[50] absolute and relative age controls is required to provide age–depth models for sediment cores that extend beyond the radiocarbon dating range ($\sim 40\,000$ years). If available, further

**Table 1.** Palaeoclimate parameters in the PalMod 130k marine data synthesis.

| Parameter |
| --- |
| Benthic foraminifera $\delta^{18}$O and $\delta^{13}$C |
| Planktonic foraminifera $\delta^{18}$O and $\delta^{13}$C |
| Seawater temperature* |
| Radiocarbon |
| Carbonate content |
| Total organic carbon content |
| Biogenic silica content |

* Inferred from various proxies (foraminifera Mg/Ca, alkenones, microfossil assemblages).

proxy time series were then added, thus ensuring a common chronology among all proxy time series measured on the same sediment core. [55]

We selected palaeoclimate parameters to synthesise the following discussion with climate modellers within the PalMod project. The high-priority selection includes both physically and biogeochemically relevant parameters, of which some are based on measurements that can be com- [60] pared with climate model output using (forward proxy) models (e.g. benthic $\delta^{18}$O) and others represent inferred parameters that can be compared with model output more directly but for which proxy models are still in their infancy (e.g. temperature based on foraminifera Mg/Ca). Also considered in [65] parameter selection was the expected spatial and temporal coverage of data availability as well as the existence of previous data products. The high-priority parameters for which data are presented here are listed in Table 1. If available, raw data were synthesised and in cases where raw data were not [70] available and it was possible to derive the raw data from the inferred palaeoclimate data, raw data were back-calculated. Raw data time series obtained in this way are flagged with a note describing the calculation.

We note that our approach of first building the strati- [75] graphic framework based on radiocarbon dates and benthic foraminifera $\delta^{18}$O means that the synthesis is not necessarily comprehensive as it does not include time series where one of the parameters of interest has been measured but where the components of the stratigraphic framework are not available. [80] However, at this stage, we opted to include only sediment cores where an age modelling strategy that is consistent and comparable across the entire data product could be achieved.

### 2.1 Data discovery

In principle, data synthesis can proceed by expansion or re- [85] duction (Fig. 1). The first, more traditional, approach relies on expert knowledge of what data are available and/or on a systematic literature search. In this approach the synthesis grows by including more data until sufficient data that meet inclusion criteria are compiled. In this way, a lot of time is [90]

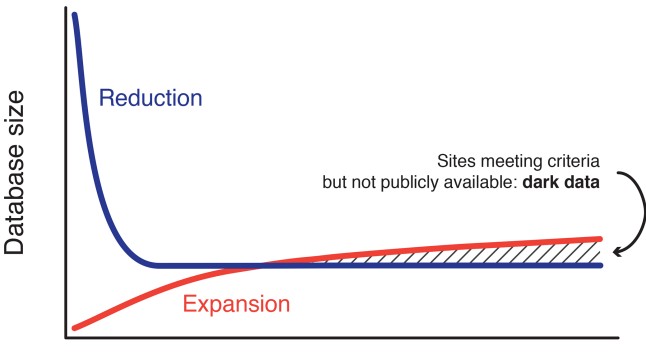

**Figure 1.** Data synthesis approaches. In the expansion approach the database size increases slowly as records are added. The database size follows an opposite pathway using the reduction approach and reaches a stable size more quickly, with less effort. Since the expansion approach is not restricted to data that are available in the public domain, this approach may lead to a database that includes data that are not publicly available (dark data). The reduction approach on the other hand is arguably more objective, can be automated and is therefore more efficient. This approach also encourages good data stewardship.

spent on discovering and retrieving datasets, and it is possible that valuable, but less exposed, data are missed. On the other hand, this approach has a chance of uncovering dark data that are not publicly available (Fig. 1).

The second approach starts from a large and crude synthesis of data from public sources and proceeds by weeding out data that do not meet criteria for inclusion in the data product. This approach faces different challenges: making sure that the initial bulk database is comprehensive (efficient data mining) and assuring that the data filtering is efficient (fast and accurate). In contrast to the expansion approach, this reduction approach cannot discover dark data. However, it is more objective (less reliant on expert knowledge), can be automated more easily and focuses on data that are already in the public domain so that no time is lost to finding data that ultimately prove unavailable. This second approach also rewards and encourages good data stewardship.

In theory, both approaches can lead to a similarly sized and exhaustive synthesis, but they differ in the allocation of effort (Fig. 1). In practice both approaches are often combined, especially towards the end of a synthesis project, when the data product is benchmarked against existing syntheses.

## 2.2   Synthesis

### 2.2.1   Initial synthesis

We followed the reduction approach and used a semi-automated pipeline to compile data from public sources. Keywords (Supplement) were used to make lists of URLs of potentially relevant data on https://pangaea.de TS3, and the

linked files were then downloaded in bulk ($n = 108\,239$). A slightly different approach was followed for the NCEI archive. Here, all files that were machine-readable at the time of download (September 2016, $n = 1925$) were obtained from the FTP CE3 server (ftp://ftp.ncdc.noaa.gov/pub/ data/paleo/paleocean//sediment_files/complete TS4). Custom scripts in R were used to put all data in a common format and merge time series that could be unambiguously assigned to the same core (based on name and $x$, $y$, $z$ position). This resulted in a mixture of records that were merged to the same core and those that could not be, either because there was only one data file for the core or because of ambiguous labelling. We refer to the locations of these records as "sites". In order to facilitate the analysis (filtering) of the sites, a uniform attribution of the various parameter names had to be developed. Because no standardised names exist for palaeoclimate parameters, the uniform attribution required the development of attribution libraries for each desired palaeoclimate parameter. The initial synthesis contained time series from 38 511 sites.

### 2.2.2   Data reduction and standardisation of ontologies

The initial synthesis was reduced by removing non-marine sites (using elevation flag) and further constrained by only considering sites where at least one data point of any of the parameters measured in that core fell within the target time frame (disambiguating age units in the synonym library of the category "age") and the site had benthic oxygen isotope data. This resulted in 781 sites. At this stage, no criteria for length or resolution were applied but we prioritised processing time series that we estimated to contain at least 50 data points within the 130 000-year timeframe. Further data processing started with dereplication of the selected sites. This was necessary because no standards exist for the naming of cores and the repositories store data with different renditions of the same core name, sometimes even associated with erroneous geographic coordinates. This process was carried out manually and proceeded by constructing a list of disambiguated sites through sequential one-by-one comparison. Where a strict synonym was found (different labels for the same core but the same data), only unique data were retained. At this stage, disambiguation of site names was only performed for sites that had at least benthic $\delta^{18}$O, so time series of other parameters, which were associated with inconsistent core labels, could have been missed in the synthesis. However, those sites are contained in the initial bulk synthesis and are hence not lost but will be salvaged in updates of the data product (see Sect. 7).

Further steps required a manual standardisation of the names of the parameters and their attributes (such as the species name that was analysed for oxygen isotopes). This was accomplished by deciding on a final, uniform list of parameters and associated metadata and their possible values. Original parameter names were preserved to allow for cross-

Earth Syst. Sci. Data, 12, 1–25, 2020                                                         www.earth-syst-sci-data.net/12/1/2020/

checking. By metadata, we refer to aspects of the individual parameters that were deemed essential to facilitate a meaningful analysis in a palaeoclimatic context, considering potential sources of uncertainty, such as species of foraminifera analysed or the calibration equation used for palaeotemperature estimates. The list of metadata is provided in Table 2. The standardisation was accompanied by further dereplication of individual time series that were already associated with the same site name but archived more than once.

### 2.2.3 Metadata and chronology

Subsequently, as far as possible, metadata values were added manually when missing, often by scraping the information from the original publication. Next, all time series from a single site (core) were put on a common depth scale to allow for age modelling. Data that could not be put on a depth scale were excluded from the synthesis. This was the case where parameter values were recorded only against age and where no other data file was available that allowed unambiguous reassignment to depth. Ambiguity also resulted from the use of multiple (composite) depth scales for the same archive. Finally, chronological data (all absolute markers, including radiocarbon dates and associated metadata) were manually added, where necessary also by consulting the original publications. Throughout the process, publication information (digital object identifier (DOI) or, if not available, full bibliographic details) and the data source (URL and/or DOI) were preserved in order to trace the source of the data. This applies both to the source of the individual data files from repositories and to the sources of the metadata and chronological data.

### 2.3 Age modelling

Whereas the initial steps of data discovery and synthesis could rely on published chronology, analysis of the complete dataset will require the development of a common chronological framework. This framework must be constructed in a way that not only allows for a consistent method of assignment of ages to depths within each core but also allows for the consistent and quantitative assessment of age uncertainty. To this end, we follow an approach that combines absolute ages (radiocarbon ages, tephra layers and palaeomagnetic events) with $\delta^{18}O$ stratigraphy. As a result, our age models may differ from those reported in the original publication(s). This is not to state that the updated age models are better (constrained), but they are constructed in a way that allows for applicability and consistency across the synthesis. The consistent approach allows for an assessment of age uncertainty jointly for all records by a Bayesian approach, generating ensembles of sedimentation histories consistent with the available age control points for each core, allowing for uncertainty estimates at each depth by considering the distribution of ages given by the ensemble.

With respect to the reporting of the chronology, we follow a transparent approach, preserving the initial age model and providing the new age models as well as all information needed to revise or update the new age models. In the final step, the age information from absolute ages and $\delta^{18}O$ tie points was combined and the age model and its uncertainty was assessed in a Bayesian framework using "bacon" (Blaauw and Christen, 2011). The entire age modelling routine was carried out in PaleoDataView (PDV; Langner and Mulitza, 2019).

To ensure a common chronological framework for all time series in the synthesis, radiocarbon ages were recalibrated using the IntCal13 curve (Reimer et al., 2013). Since reservoir ages vary in space as well as in time, we used reservoir age estimates based on a comprehensive ocean general circulation model (Butzin et al., 2017) to account for this variability in a physically plausible way. To derive the reservoir age and uncertainty for a measured radiocarbon age, PDV (i) extracts all modelled radiocarbon ages from the nearest grid cell in the modelled dataset, (ii) finds all modelled radiocarbon ages that are possible within the error of the measured radiocarbon age, and (iii) takes the mean and the standard deviation of all corresponding reservoir ages to correct for the measured radiocarbon age. By definition, this approach cannot account for processes affecting the reservoir ages on subgrid spatial scales. Given the relatively coarse resolution of the model, this means that processes such as upwelling are not fully accounted for. In addition, no modelled reservoir age data are available for the Mediterranean and Red seas; we use the reservoir ages reported by the authors of the original publication and an assumed uncertainty of 100 years for these basins (five sites). Absolute ages based on North Atlantic tephra layers and palaeomagnetic events were updated and harmonised using Svensson et al. (2008).

In addition, and beyond the $^{14}C$ dating realm ($\sim$ 40 000 years), the age models rely on manual tuning of the benthic foraminifera $\delta^{18}O$ time series from each core to regional benthic foraminifera $\delta^{18}O$ stacks (Lisiecki and Stern, 2016). Stable-isotope stratigraphy in theory provides a range of events to correlate; however, in order to not inflate confidence in the tuned age models and to ensure comparability between different cores, in our approach, the tuning was carried out as far as possible by only matching the position of marine isotope stage boundaries. We updated the age–depth models only for the 0–130 000 years time frame of this synthesis, but data and original age models extending beyond 130 000 years are preserved in the data product. To obtain uncertainty for the age control points obtained by $\delta^{18}O$ tuning, we used the chronological uncertainty in the $\delta^{18}O$ stacks, as reported by Lisiecki and Stern (2016). Additional uncertainty associated with the identification of the control points in the individual records or with the assumption on synchronicity was ignored, as these are difficult, if not impossible, to quantify.

**Table 2.** Metadata terms.

| Name | Description |
| --- | --- |
| ParameterOriginal* TS5 | Original parameter name as in data repository |
| Parameter* | Standardised parameter name (see Table 4) |
| ParameterType* | Parameter type, measured or inferred |
| ParameterUnit* | |
| ParameterAnalyticalError | Error based on repeat measurements of standards |
| ParameterReproducibility | Error based on repeat measurements of samples |
| Instrument | |
| Laboratory | |
| SampleThickness_cm | |
| Material* | Measurement material or parameter on which inferred parameter is based |
| Species | |
| Nshells | |
| SizeFraction_microm | |
| Notes | |
| RecordingSeason | |
| RecordingDepth | |
| EquilibriumOffset | |
| CalibrationEquation | |
| CalibrationUncertainty | |
| CalibrationDOI | |
| TransferFunctionTrainingSet | |
| TransferFunctionUncertainty | |
| TransferFunctionDOI | |
| PublicationDOI* | |
| Authors | |
| PublicationTitle | |
| Journal | |
| Year | |
| Volume | |
| Issue | |
| Pages | |
| ReportNumber | |
| DataDOI | |
| DataLink* | |
| RetrievalNumber | For internal use only |

## 3 Notes on palaeoclimate proxies in marine-sediment archives and metadata

This synthesis contains climate-sensitive proxy data based on measurements using various biological sensors. It is not the intention here to provide a full overview of marine palaeoclimate proxies and their uncertainties (for this, see for example Hillaire-Marcel and De Vernal, 2007; Moffa-Sánchez et al., 2019), but the fact that the proxies are based on biological sensors means that they are affected by different ecological bias in addition to observational noise. Basic knowledge of the recording system is therefore essential for the interpretation of the data and may aid in explaining differences between proxies for the same climate parameter. These considerations were also essential to choosing the range of metadata to be recorded alongside each palaeoclimatic parameter to allow for a proxy-specific assessment of uncertainty.

Foraminifera are among the most widely used proxy sensors in palaeoceanography. They are unicellular marine zooplankton. The species used here all build a calcite skeleton that is preserved in the sediment. Foraminifera can be divided into two main groups: benthic foraminifera living at the seafloor level or at shallow depth in the sediment and planktonic foraminifera living in the upper hundreds of metres of the ocean. In the data product, proxies measured on these two groups are clearly distinguished by a benthic or planktonic prefix. The chemical composition of foraminifera reflects environmental conditions of the seawater that the organisms calcified in. For the purpose of this data product, the parameters of interest are stable-oxygen-isotope, stable-carbon-isotope and Mg/Ca ratios. Stable-oxygen-isotope ratios in foraminifera calcite reflect a combination of temperature and $\delta^{18}$O of seawater (Urey, 1948), which is in turn related to ice volume and salinity. Species-specific calibrations

exist to quantitatively link $\delta^{18}O_{foraminifera}$ and $\delta^{18}O_{seawater}$ to temperature (e.g. Marchitto et al., 2014; Bemis et al., 1998). Stable-carbon-isotope ratios ($\delta^{13}C$) reflect the $\delta^{13}C$ of the dissolved inorganic carbon in seawater. In particular the benthic foraminifera species *Cibicidoides wuellerstorfi* generally incorporates $\delta^{13}C_{DIC}$ without a biological offset and can serve as a tracer of bottom-water $\delta^{13}C_{DIC}$ which is commonly used as non-passive circulation tracer (Curry and Oppo, 2005). The $\delta^{13}C$ of other benthic foraminifera species is generally not indicative of bottom-water $\delta^{13}C_{DIC}$, and the $\delta^{13}C$ of planktonic foraminifera is also influenced by temperature and carbonate ion concentration, rendering interpretation complicated (Spero et al., 1997).

The Mg/Ca ratio in foraminifera calcite can be used to infer calcification temperature and, in combination with $\delta^{18}O_{foraminifera}$, the $\delta^{18}O_{seawater}$ (Elderfield and Ganssen, 2000). Similar to stable oxygen isotopes, species-specific calibrations exist to quantitatively reconstruct past temperature from Mg/Ca ratios (Anand et al., 2003; Lear et al., 2002), and whenever indicated in the original publication, the calibration is included in the metadata. Carbonate system parameters and salinity have a secondary influence on Mg/Ca ratios in foraminifera calcite (Gray et al., 2018). Whereas benthic foraminifera live in a generally stable environment, the near-sea-surface habitat of planktonic foraminifera shows large seasonal and vertical gradients. Species-specific seasonal and/or depth habitat preferences may therefore leave a considerable imprint on the proxy signal contained in their shells (Jonkers and Kučera, 2017; Mix, 1987). For all proxies based on foraminifera, it is relevant to record the species as well as the number of individuals that were pooled for geochemical analysis. The latter is because the short lifespan and variable habitat of foraminifera species cause large variability among individuals. Planktonic foraminifera shell size may for several reasons also affect their chemistry (Jonkers et al., 2013; Friedrich et al., 2012) as well as their assemblage composition (Al-Sabouni et al., 2007). Therefore, the size fraction of the analysed shells was included in the metadata whenever this information was available.

Besides planktonic foraminifera Mg/Ca ratios, the $U^{K'}37$ unsaturation index can provide information about near-sea-surface temperature. The $U^{K'}37$ index is based on the relative degree of unsaturation of $C_{37}$ alkenones, which is linearly related to temperature (Prahl et al., 1988). Alkenones are produced by coccolithophores, marine phytoplankton living in the photic zone. The production of alkenones is in many regions not constant during the year, thus potentially causing a seasonal recording bias in the $U^{K'}37$ temperature proxy (Rosell-Melé and Prahl, 2013). Several calibrations exist that relate the index to sea surface temperature, and if the calibration was mentioned in the original publication, it was preserved in the metadata.

A large proportion of the temperature estimates in this data product are based on microfossil (planktonic foraminifera, diatoms, Radiolaria, dinoflagellate cysts) assemblages. These reconstructions are based on a statistical relationship between species assemblages and temperature (Imbrie and Kipp, 1971). In theory, microfossil assemblages can be used to reconstruct temperatures of different seasons or different environmental parameters from the same assemblage. However, it is not always clear that such reconstructions are truly independent (Telford and Birks, 2011). Several different methods exist to relate fossil assemblages to temperature, and researchers often apply more than a single method in their reconstructions to increase confidence (Kucera et al., 2005). When available, these different reconstructions are included in the data product.

The bulk sediment data ($CaCO_3$, TOC and BSi) form a category of their own. They are not proxies in the strict sense but properties of the sediment that reflect a combination of export productivity, sedimentation and preservation. However, they can provide crucial information about the ocean–climate system, in particular about biogeochemical cycles (Cartapanis et al., 2016). With the advent of explicit sediment modules in climate models (Heinze et al., 1999; Kurahashi-Nakamura et al., 2020), sediment composition can also be directly compared with model output and potentially provide additional constraints on the simulations.

## 4 Structure of the database

Following the data synthesis strategy outlined above, we generated a first data product for time series of eight parameters in sediment cores with radiocarbon and benthic $\delta^{18}O$ stratigraphy. Following the logic of our approach, the synthesis is organised by the physical object from which the records were extracted (cores), here called site, to account for the inclusion of records from spliced cores.

Each site in the data product has information on seven different themes (Fig. 2):

1. Geographic data contain the site name, latitude and longitude (in decimal degrees N and E), elevation or water depth (in metres), and possible notes that are relevant to the site or core as a whole. All fields except notes are essential and always included.

2. Metadata include the original parameter name as given in the online data file, a standardised parameter name, parameter type (measured or inferred), unit, an estimate of analytical error as determined from repeat measurements of a standard and an estimate of reproducibility as determined by repeat measurements on samples. For measured parameters, information on the instrument and laboratory is given. All metadata terms are listed in Table 2, and an overview of the standardised parameter names is provided in Table 3.

3. Chronology data contain raw data on absolute age control points used for age modelling. This includes not

only depth, radiocarbon ages including their uncertainty, dated material and laboratory codes but also calendar ages of tephra layers and palaeomagnetic events. Age control points not used in the age model by the authors of the original publication(s) are indicated, and if available, the source (DOI or URL) of the data is shown in addition to the original publication DOI. A complete list of chronology data terms is given in Table 4.

4. The actual time series data are provided on a common depth scale. Original age models are preserved alongside the data time series as they may differ for different time series from the same site. The data also contain information on the sample number or label (mainly for DSDP, ODP and IODP CE4 cores) for spliced records and sample-specific notes.

5. The revised age model contains ages for depths bracketed by age control points (absolute and relative). Mean and median ages are given as well as an uncertainty range (2.5th and 97.5th percentiles) based on the full suite of age model ensembles. No attempts were made to extrapolate the age models beyond the tie points in the 130 000-year time frame, so original age models may extend to either side.

6. The bacon data contain all information to reproduce the revised age model. Besides the $^{14}$C and absolute age control points, this includes the tie points for the alignment, the alignment target and all parameters used to construct the age–depth model.

7. Age ensembles are provided for further assessment of chronological uncertainty. In order to keep file sizes manageable, 1000 randomly selected age model ensembles are preserved.

## 5   PalMod 130k marine palaeoclimate data synthesis v1.0.0 contents

The data product contains 896 time series of the palaeoclimate parameters listed in Table 1 from 143 sites (Table 5). By design all sites have both benthic stable-oxygen-isotope and radiocarbon data. The majority of the sites are close to the continents and in the Northern Hemisphere, with a concentration in the North Atlantic Ocean (Fig. 3). This reflects both research attention as well as the challenges of obtaining sediment cores with high accumulation rates and well-preserved foraminifera. The effect of research focus is also visible in the temporal coverage of the time series, where every parameter is characterised by a clear maximum around the last glacial termination ca. 15 ka (Fig. 4). The median resolution of the time series varies by 2 orders of magnitude but is generally better than one sample per 1000 years and fairly similar among the different parameters (Fig. 5). The updated

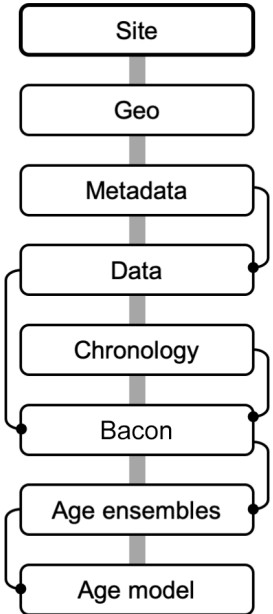

**Figure 2.** Structure of the PalMod 130k marine palaeoclimate data synthesis. Each file in the database contains information on seven different themes on a single site (sediment core). Links between different themes are indicated.

age models are based on chronological control points (tie points) from radiocarbon dating and absolutely dated layers (using tephra and/or palaeomagnetic event stratigraphy) as well as alignment to the regional benthic $\delta^{18}$O stacks. The majority of the time series has a chronological control point at least every 5000 years (Fig. 6). Taken together, the coverage in space, time and across parameters indicates that the PalMod marine palaeoclimate data product allows for analysis of palaeoclimate on a supra-regional scale over the entire 130 000-year time frame.

This data product builds upon previous syntheses. Virtually all of the sites are also part of the benthic foraminifera $\delta^{18}$O and $\delta^{13}$C compilations of Lisiecki and Stern (2016) and Peterson and Lisiecki (2018). Our synthesis, however, also includes data on other palaeoclimate parameters and contains more metadata and information on the age–depth models. Some of the planktonic foraminifera $\delta^{18}$O and Mg/Ca time series in the PalMod 130k data product are also included in the Iso2k synthesis effort (Konecky et al., 2018), and a number of sea surface temperature time series are also part of the forthcoming Temperature12k synthesis (Kaufman et al., 2020).

## 6   Data formats and access

We provide the data products in three different formats in order to facilitate access and analysis using different software and across operating systems. Given the structure of the formats, each representation is slightly different in its level of

**Table 3.** Standardised parameter names.

| Name | Description |
|------|-------------|
| benthic.d18O | Benthic foraminifera $\delta^{18}O$ |
| benthic.d13C | Benthic foraminifera $\delta^{13}C$ |
| planktonic.d18O | Planktonic foraminifera $\delta^{18}O$ |
| planktonic.d13C | Planktonic foraminifera $\delta^{13}C$ |
| surface.temp | Inferred (near-)sea-surface temperature (based on microfossils, planktonic foraminifera Mg/Ca, $U^{K'}37$) |
| deep.temp | Inferred bottom-water temperature (based on benthic foraminifera Mg/Ca) |
| CaCO3 | Calcium carbonate content |
| TOC | Total organic carbon content |
| BSi | Biogenic silica content |
| DBD | Dry bulk density |
| IRD | Ice-rafted detritus |
| planktonic.MgCa | Planktonic foraminifera Mg/Ca ratio |
| benthic.MgCa | Benthic foraminifera Mg/Ca ratio |
| UK37 | $U^K37$ ratio (rare cases where this is not $U^{K'}37$ mentioned in notes) |
| C37.concentration | Alkenone concentration |

**Table 4.** Chronology terms.

| Name | Description |
|------|-------------|
| ChronType* | Type of absolute chronology tie point ($^{14}C$, tephra, palaeomag) |
| ChronDepthTop_cm | |
| ChronDepthBottom_cm | |
| ChronDepthMid_cm* | |
| ChronSampleThickness_cm | |
| ChronAge_kaBP* CE5 | Age of non-$^{14}C$ tie point (tephra, palaeomag) |
| ChronAgeError_ka* | Age error of non-$^{14}C$ tie point (tephra, palaeomag) |
| ChronDatedMaterial | |
| ChronDatedSpecies | |
| ChronNshellsDated | |
| Chron14CLabcode | |
| ChronAge14C_kaBP* | |
| ChronAge14CError_ka* | |
| ChronAge14CErrorUp_ka | |
| ChronAge14CErrorDown_ka | |
| ChronReservoirAge_ka* | |
| ChronReservoirAgeError_ka* | |
| ChronCalibCurve | |
| ChronCalibAge14C_kaBP | Calibrated $^{14}C$ age |
| ChronCalibAge14C1sigLo_ka | |
| ChronCalibAge14C1sigUp_ka | |
| ChronAgemodelMethod | |
| ChronAgeRejected | |
| ChronNotes | |
| ChronSource | |
| ChronDOI* | |

\* Essential terms.

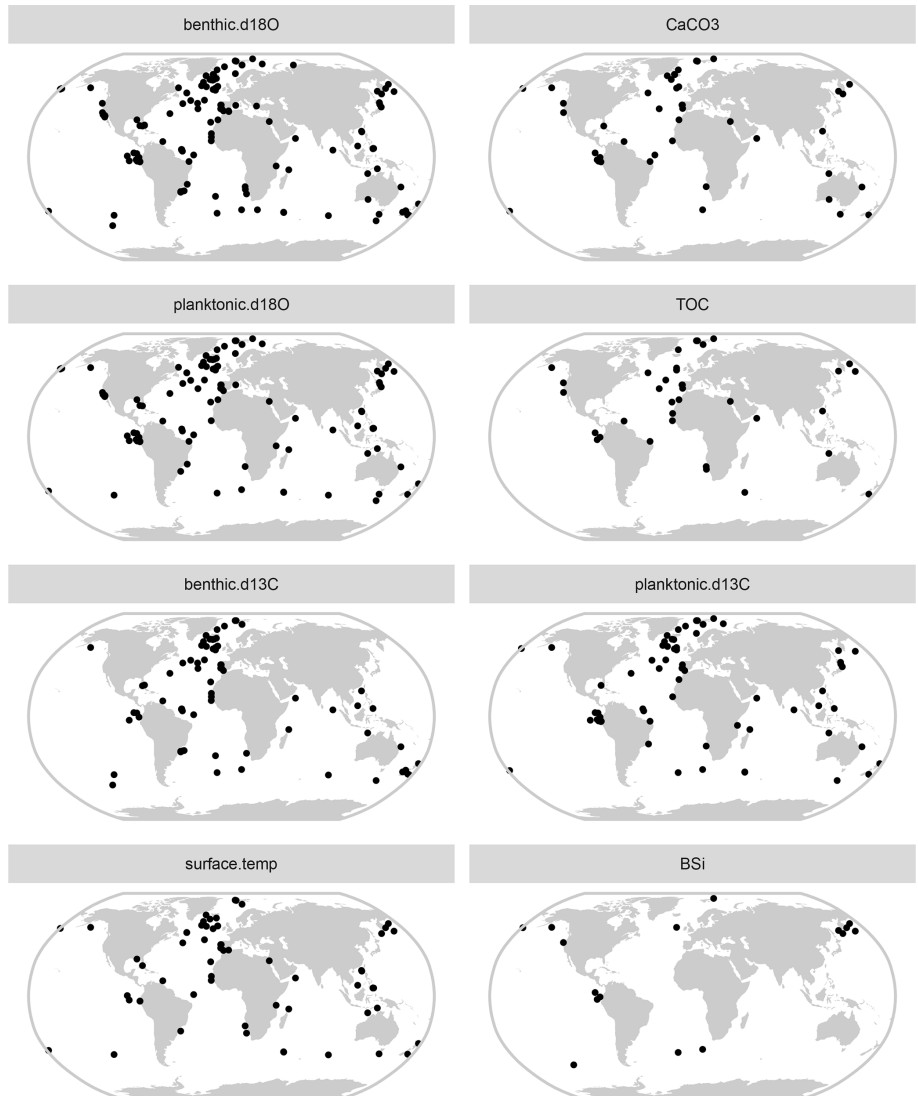

**Figure 3.** Spatial distribution of sites in version 1.0.0 of the PalMod 130k marine palaeoclimate data synthesis. The distribution of the sites reflects research effort and the possibility of obtaining sediment cores containing well-preserved foraminifera and is hence skewed towards the Northern Hemisphere (Atlantic) and the continental margins. Since no explicit search was carried out for parameters other than benthic foraminifera $\delta^{18}$O, the distribution of the other parameters is restricted to sites that have benthic foraminifera $\delta^{18}$O. For benthic foraminifera $\delta^{13}$C, only sites with data based on the genus *Cibicidoides* are shown.

metadata detail and the way metadata and data are stored. The differences are described below.

– Since the data product was built using R, the data product is presented in R-readable RDS files that contain for each site a list with data for each theme (Sect. 4). This is the format that is most complete, yet in the interest of memory space it preserves a random selection of 1000 age models from the larger ensemble produced using bacon. In this format, all data and metadata for each site are contained within a single file. Sample scripts (https://github.com/lukasjonkers/PALMODutils TS6) allow the user to extract a quick overview of the contents

of the data product similar to Table 5 but with additional information on the temporal range, resolution and age control of the time series. Additional code is available to query the data product by parameter, parameter detail, sensor species, temporal range, resolution and age control.

– The data product is also provided in the LiPD format, which is built around JSON-LD and CSV formats and is widely readable across different platforms. As with RDS, all data for each site are presented in a single file. Utilities to interact with LiPD files in R, Matlab and

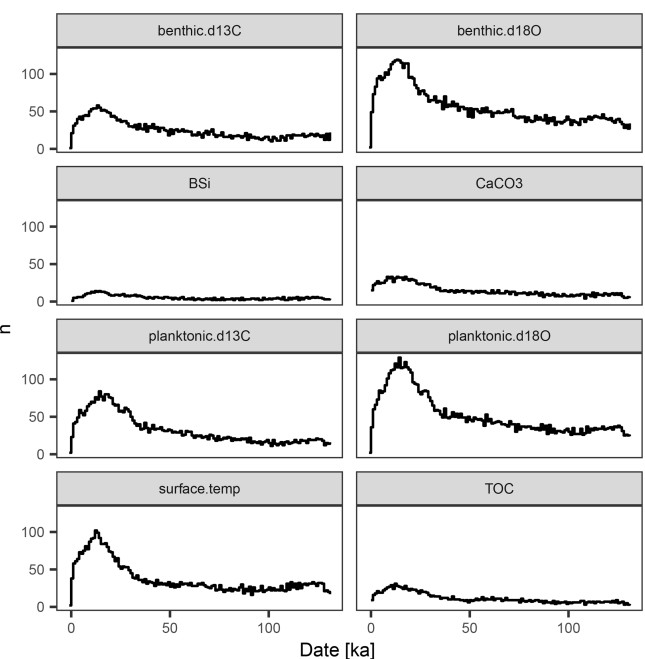

**Figure 4.** Temporal distribution of time series in version 1.0.0 of the PalMod 130k marine palaeoclimate data synthesis. The number of time series ($y$ axis) is counted per 1000-year bin. The temporal availability of all parameters shows a clear maximum around 15 ka, reflecting the research focus on the last glacial termination. For benthic foraminifera $\delta^{13}$C, only time series with data based on the genus *Cibicidoides* are shown.

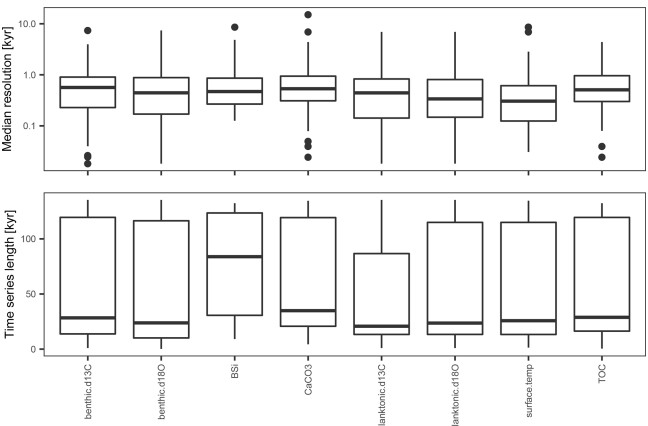

**Figure 5.** Median resolution of the time series in version 1.0.0 of the PalMod 130k marine palaeoclimate data synthesis. Box-and-whisker plots show the spread of resolution per parameter. For benthic foraminifera $\delta^{13}$C the resolution data are restricted to time series containing data measured on shells of the genus *Cibicidoides*. For all parameters the median resolution is more than one data point per 1000 years.

Python are available at https://github.com/nickmckay/ LiPD-utilities TS7.

– Finally, the data are also provided in netCDF format in a way that allows reading in PDV format. This means that a single site has separate files for each individual palaeoclimate parameter as well as for the age model. In this format, some metadata are stored as concatenated strings rather than easily searchable attributes. The netCDF format allows however for the storage of the full suite of age model ensembles without excessive file sizes. The PDV software to read and process the data can be downloaded at https://www.marum.de/ en/Stefan-Mulitza/PaleoDataView.html TS8.

The data product is available for download at https://doi.org/10.1594/PANGAEA.908831 (Jonkers et al., 2019). We encourage users of the data product to also cite the primary source of the data when using (individual time series of) this product.

## 7 Future plans and versioning

To increase the spatio-temporal coverage over the entire 130 000-year time frame of the database, updates of this data product will first aim for quantitative growth of the database by adding more time series with chronological control based on benthic foraminifera $\delta^{18}$O and absolute age control points other than $^{14}$C. If available, these updates will also include the parameters listed in Table 1. They will be named using the counter following the first decimal separator. The structure of the data product is designed to be flexible, allowing for the addition of different metadata fields and parameters. Further updates that include new parameters and/or require a new age modelling approach (i.e. no benthic $\delta^{18}$O alignment) will be named using the counter before the first decimal separator. Any updates to add or correct (meta)data to an existing version that do not increase the number of sites will be indicated using a counter following the second decimal separator.

## 8 Data availability

An overview of all datasets used in this synthesis, including URLs to the data, can be found in Table 5 (https://doi.org/10.5281/zenodo.3739019 TS9). The PalMod 130k marine palaeoclimate data product can be downloaded in R, LiPD and netCDF format at https://doi.org/10.1594/PANGAEA.908831 (Jonkers et al., 2019). The data can also be visualised and downloaded in LiPD and CSV formats at http://lipdverse.org/PalMod/ current_version/.

## 9 Lessons learned: recommendations for data archiving

Data reuse and sharing are both made easier when data are archived in a standardised manner. Even though a large num-

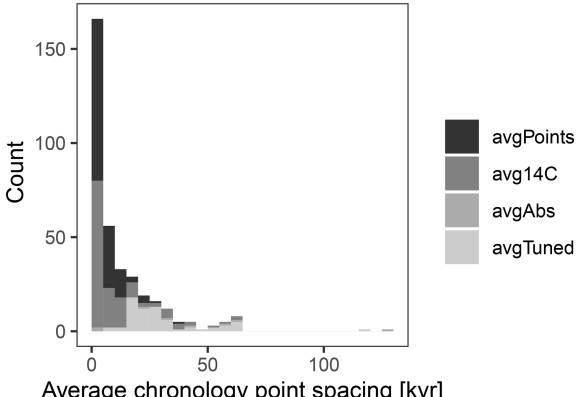

**Figure 6.** Average temporal spacing between chronology tie points (avgPoints). Tie points are also split into $^{14}$C, absolute (tephra or palaeomagnetics) and tuned. The majority of the time series in the synthesis has a spacing of chronology tie points that is below 5000 years.

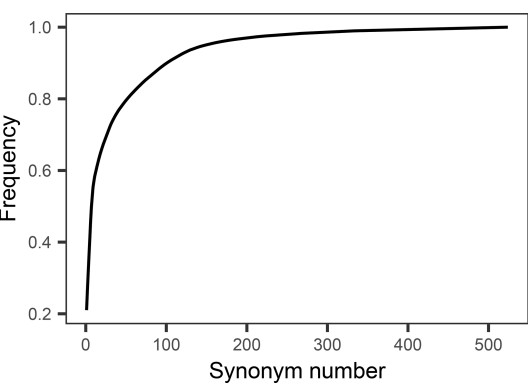

**Figure 7.** The problem of the absence of standardisation in parameter names. The cumulative frequency of synonyms for seawater temperature in our initial database (Sect. 2.2.1), showing that there are over 500 different names for the same parameter and that many of these are unique.

ber of palaeoceanographic data are publicly available, metadata to facilitate interpretation of the raw or inferred palaeodata are often not, or only partially, made available and need to be obtained from the original publication. Synthesis efforts
therefore still require a lot of time and effort to find, compile and standardise data and metadata. Only recently has the palaeoclimate community started to discuss data-archiving standards (Khider et al., 2019). However, implementation of the proposed Paleoclimate Community reporTing Standard
(PaCTS 1.0) will only affect new uploads to public repositories, and data already available (legacy data) are likely only going to be made compliant with the PaCTS through dedicated synthesis efforts. Below we list some of the main issues that we encountered during data synthesis. Our aims
with mentioning these are to raise awareness of how the lack of standardisation affects data synthesis and thereby to encourage best practice in data reporting. We encourage researchers and also reviewers to treat data handling not as an afterthought but as an integral part of their study. After
all, compared to generating the data, data handling is not a time-consuming task. Time spent on proper documenting and archiving is not wasted as it facilitates reuse of data and enables scientific progress in our field.

*Disambiguate core names.* An apparently trivial, but sur-
25 prisingly common, first-order issue is that core names are inconsistently archived. Different names for the same core arise not only from differences in hyphenation; truncation of (long) names; and minor variations in the same name that can, with expert knowledge, be linked but also from the use
of altogether different names for the same core, e.g. reflecting differences in the labelling during an expedition and in the repository. This naming confusion renders it difficult to combine datasets from the same core, especially in an automated way, and to assess the uniqueness of time series from
the same core for dereplication. We recommend using the

full name as indicated in the cruise report where the core was first described.

*Standardise vocabularies.* Even though a vast number of palaeoclimate data are available in public repositories, the lack of standardisation of parameter vocabularies hinders ef- 40 ficient data processing. This problem is clearly illustrated by the fact that, for this synthesis, long synonym lists needed to be generated in order to group parameters. Each parameter in this synthesis had tens to hundreds of different names, of which many were unique (Fig. 7). This issue can be partly ad- 45 dressed by a consistent separation of parameter and attribute names (e.g. parameter $\delta^{18}$O, species *G. bulloides*, instead of a single parameter "d18OGbul"), but even that calls for a standardisation of parameters and attributes.

*Report sampling depth.* All data reported here are based on 50 measurements of discrete samples from a specific depth interval in a given core. Therefore, the synthesis requires information on the position of each sample. Since ages of the samples are always estimates and may differ among studies of the same archive, unambiguous information on sample depth 55 is essential to reproduce and update the time series. Despite this, many studies fail to report sample depth and instead report only age. This problem is worse for spliced records that rely on a composite depth scale. Splicing approaches are often opaque, and original sample depths, or sample codes 60 that identify unique samples, are not always available. We recommend therefore that sample depth should be essential for palaeodata time series from (marine) sediments and that sample labels are archived for spliced records. This includes (I)OPD or DSDP sample labels (in full) or IGSN (if avail- 65 able).

*Publish raw data.* To ensure the reproducibility of inferred parameters (in this synthesis only temperature) and to ensure the harmonisation or updating of the calibration, the raw measured data are needed. Provided that the calibration is 70

known, measured data can in some cases be calculated from the inferred data, but this is impossible for data based on microfossil transfer functions. Related to this is unclear information about the calibration that was used, particularly if the publication describing the calibration includes multiple different equations.

*Include metadata.* To assess ecological imprints on proxy signals (recording bias), temperature estimates as well as oxygen and carbon isotope data based on planktonic foraminifera also require that key metadata, such as species name, are archived in a standardised way. This is not universally carried out, and for example the species information is often only available in the original publication. Additional information to assess the uncertainty in proxy measurements, such as foraminifera shell size, the sample size (e.g. number of shells, concentration of alkenones) or reproducibility of repeat measurements, was often not available from the paper or from the archived data, and we encourage the archiving of such data in a standardised way. A similar issue applies to chronological data. Thanks to a longer history of reporting standards, radiocarbon (Stuiver and Polach, 1977) (meta)data are often rather complete. However, this information is often not included alongside the digitally available data, and for this synthesis a large proportion of the radiocarbon (meta)data had to be scraped from the literature. In our age modelling approach, we took reservoir age uncertainty into account, using data derived from the modelled reservoir ages (see Sect. 2.3). Alternative approaches are hindered by the fact that reservoir age uncertainty is almost never reported.

*Avoid redundancy.* A considerable amount of time was spent on the dereplication of time series of the same parameter from the same core that were archived multiple times. Repeat archiving happens when data are reused or, less commonly, updated. The dereplication task is not made easier by (incomplete or inconsistent) metadata reporting and can be avoided through better linking of existing datasets when, instead of re-uploading the data, the DOI or URL of the original data is provided.

*Help rescue dark data.* This synthesis is based on data that are publicly available, yet many or some palaeoceanographic time series are not archived in public repositories. Even though the proportion of this so-called dark data is unknown, it likely affects every branch of palaeoceanography, and as a result palaeoceanographical data syntheses cannot be exhaustive. This problem is clearly exacerbated for syntheses relying on automated data mining. There are several shades of dark data, each requiring their own approach to retrieve them and make them available. Some data are only partially available, for instance datasets that lack sample depths. Such data only require additional data to make them reusable. These additional data can sometimes be calculated or obtained from cross-referencing different data files but in many cases will need to be retrieved from the data producers. Other datasets, in particular those from before the dig-

ital age, are presented in tables in the original publications. Progress has been made with digitising those datasets (especially in PANGAEA), but this work is not finished, and more effort is needed to make this data available to the community. There are also data that are used in publications but are not made available in any way (print or digital). This third shade of data can so far only be obtained from the original data producers or authors of the original publication or, if this proves impossible, needs to be digitised from graphs. Digitisation inevitably leads to a loss of accuracy of the data, and a dataset retrieved in this way should be flagged. A final category of dark data consists of data that are not part of a publication. Such datasets can be made publicly available and be associated with a DOI to ensure traceability. To reward data sharing, the use of data citations needs to be encouraged and data citations should be included in the evaluation of a researcher's impact.

Table 5 (online only at https://doi.org/10.5281/zenodo.3739019 TS10: Palaeoclimate time series in the PalMod 130k marine palaeoclimate data synthesis v1.0.1). This table lists the site names and locations and parameters including additional information as well as the source of the data and the original publications where the data were presented.

**Supplement.** The supplement related to this article is available online at: https://doi.org/10.5194/essd-12-1-2020-supplement.

**Author contributions.** LJ, MK and SM conceptualised the study; LJ, OC and MK developed the workflow; LJ designed the database structure with help from OC and MK; LJ and AS compiled (meta)data; LJ carried out quality control; LJ updated age–depth models; LJ, SM and ML ensured integration with the PDV format; LJ and NM ensured the integration with the LiPD format; MK and SM acquired funding; LJ and MK wrote the manuscript, and all authors reviewed and commented on the manuscript.

**Competing interests.** The authors declare that they have no conflict of interest.

**Special issue statement.** This article is part of the special issue "Paleoclimate data synthesis and analysis of associated uncertainty (BG/CP/ESSD inter-journal SI)". It is not associated with a conference. TS11

**Acknowledgements.** This research was supported by PalMod, the German palaeoclimate modelling initiative (http://www.palmod.de TS12). PalMod is part of the Research for Sustainable Development initiative (FONA; http://www.fona.de TS13) funded by the German Federal Ministry of Education and Research (BMBF). Olivier Cartapanis was funded by the Swiss National Science Foun-

dation (grant PP00P2-144811). We thank colleagues in working group 3 from PalMod for discussion on data and metadata requirements and Rangelys Sorrentino for help with data processing. We also thank Blanca Ausin and an anonymous reviewer for their constructive feedback on an earlier version of this manuscript.

**Financial support.** This research has been supported by the Bundesministerium für Bildung und Forschung (PalMod grant). TS14

**Review statement.** This paper was edited by David Carlson and reviewed by Blanca Ausin and one anonymous referee.

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

## Remarks from the language copy-editor

CE1    Please note the edits to the affiliations.

CE2    If "is" refers to information, then it is fine. If "is" refers to the proxies, then it should be changed to "are". Please clarify.

CE3    This abbreviation is used only once. Is it well known, or should it be defined for clarity?

CE4    The abbreviations DSDP, ODP and IODP are not defined. Are they well known, or should they be defined for clarity?

CE5    Please note that we apply the standard of using "ka" rather than "ka BP" to indicate dates in the past; ka BP is redundant because ka already implies years before present, which is defined as 1950. An exception to this is the "4.2 ka BP event". We use BP with "kyr" to give durations and with calibrated dates (e.g. 4.2 ka cal BP). This means that the chronology terms here that include kaBP conflict with the standards applied to the rest of the text. Please check the relevant changes throughout and let us know if any further adjustments need to be made.

## Remarks from the typesetter

TS1    The composition of Fig. 6 has been adjusted to our standards. Language adjustments have been made to Figs. 2, 4 and 6.

TS2    Please provide date of last access.

TS3    Please provide date of last access.

TS4    Please provide date of last access.

TS5    Please let me know which information this asterisk represents. Is it the same as for Table 4?

TS6    Please provide date of last access.

TS7    Please provide date of last access.

TS8    Please provide date of last access.

TS9    Please check DOI.

TS10    Please provide a citation and check DOI. Please also provide a placement of this information in the text. Furthermore, as this is an additional table and not part of the paper, please rename this table information (e.g. "an additional table/an additional parameter list can be found at").

TS11    Please confirm.

TS12    Please provide date of last access.

TS13    Please provide date of last access.

TS14    Please note that there is funding information given in the acknowledgements but you have not indicated any funding upon manuscript registration. Therefore, we were not able to complete the financial support statement. Please provide the missing information and double-check your acknowledgements to see whether repeated information can be removed from the acknowledgements. Thanks.

TS15    Reference not mentioned in text.

TS16    Reference not mentioned in text.

TS17    Reference not mentioned in text.

TS18    Please provide article number or page range.

TS19    Reference not mentioned in text.

TS20    Reference not mentioned in text.

TS21    Reference not mentioned in text.

TS22    Please provide article number or page range.

TS23    Reference not mentioned in text.

TS24    Reference not mentioned in text.

TS25    Reference not mentioned in text.

TS26    Please provide article number or page range.

TS27    Reference not mentioned in text.

TS28    Please provide article number or page range.

TS29    Reference not mentioned in text.

TS30    Reference not mentioned in text.

TS31    Please provide article number or page range.

TS32    Reference not mentioned in text.

TS33    Reference not mentioned in text.

TS34    Reference not mentioned in text.

TS35    Reference not mentioned in text.

TS36 Reference not mentioned in text.
TS37 Reference not mentioned in text.
TS38 Reference not mentioned in text.
TS39 Please provide article number or page range.
TS40 Reference not mentioned in text.
TS41 Reference not mentioned in text.
TS42 Reference not mentioned in text.
TS43 Please provide article number or page range.
TS44 Reference not mentioned in text.
TS45 Please provide article number or page range.
TS46 Reference not mentioned in text.
TS47 Please provide article number or page range.
TS48 Reference not mentioned in text.
TS49 Reference not mentioned in text.
TS50 Reference not mentioned in text.
TS51 Reference not mentioned in text.
TS52 Reference not mentioned in text.
TS53 Reference not mentioned in text.
TS54 Reference not mentioned in text.
TS55 Reference not mentioned in text.
TS56 Reference not mentioned in text.
TS57 Reference not mentioned in text.
TS58 Reference not mentioned in text.
TS59 Please provide article number or page range.
TS60 Reference not mentioned in text.
TS61 Reference not mentioned in text.
TS62 Reference not mentioned in text.
TS63 Reference not mentioned in text.
TS64 Please provide article number or page range.
TS65 Reference not mentioned in text.
TS66 Reference not mentioned in text.
TS67 Reference not mentioned in text.
TS68 Reference not mentioned in text.
TS69 Please provide article number or page range.
TS70 Reference not mentioned in text.
TS71 Reference not mentioned in text.
TS72 Reference not mentioned in text.
TS73 Reference not mentioned in text.
TS74 Reference not mentioned in text.
TS75 Reference not mentioned in text.
TS76 Please provide article number or page range.
TS77 Reference not mentioned in text.
TS78 Reference not mentioned in text.
TS79 Reference not mentioned in text.
TS80 Reference not mentioned in text.
TS81 Reference not mentioned in text.
TS82 Reference not mentioned in text.
TS83 Please provide article number or page range.
TS84 Reference not mentioned in text.
TS85 Please provide article number or page range.
TS86 Reference not mentioned in text.
TS87 Reference not mentioned in text.
TS88 Please provide article number or page range.
TS89 Reference not mentioned in text.
TS90 Reference not mentioned in text.

TS91    Reference not mentioned in text.
TS92    Reference not mentioned in text.
TS93    Reference not mentioned in text.
TS94    Reference not mentioned in text.
TS95    Please provide article number or page range.
TS96    Reference not mentioned in text.
TS97    Please provide article number or page range.
TS98    Reference not mentioned in text.
TS99    Reference not mentioned in text.
TS100    Reference not mentioned in text.
TS101    Reference not mentioned in text.
TS102    Reference not mentioned in text.
TS103    Reference not mentioned in text.
TS104    Reference not mentioned in text.
TS105    Reference not mentioned in text.
TS106    Reference not mentioned in text.
TS107    Reference not mentioned in text.
TS108    Please provide an initial and the full author list, if available.
TS109    Please provide an update.
TS110    Reference not mentioned in text.
TS111    Reference not mentioned in text.
TS112    Reference not mentioned in text.
TS113    Reference not mentioned in text.
TS114    Reference not mentioned in text.
TS115    Reference not mentioned in text.
TS116    Please provide volume with article number or page range.
TS117    Reference not mentioned in text.
TS118    Please provide article number or page range.
TS119    Reference not mentioned in text.
TS120    Reference not mentioned in text.
TS121    Reference not mentioned in text.
TS122    Reference not mentioned in text.
TS123    Reference not mentioned in text.
TS124    Reference not mentioned in text.
TS125    Reference not mentioned in text.
TS126    Please provide article number or page range.
TS127    Please provide article number or page range.
TS128    Reference not mentioned in text.
TS129    Please provide article number or page range.
TS130    Reference not mentioned in text.
TS131    Reference not mentioned in text.
TS132    Reference not mentioned in text.
TS133    Reference not mentioned in text.
TS134    Reference not mentioned in text.
TS135    Reference not mentioned in text.
TS136    Reference not mentioned in text.
TS137    Reference not mentioned in text.
TS138    Please note update to final revised version.
TS139    Please provide article number or page range.
TS140    Reference not mentioned in text.
TS141    Reference not mentioned in text.
TS142    Reference not mentioned in text.
TS143    Reference not mentioned in text.
TS144    Reference not mentioned in text.
TS145    Reference not mentioned in text.

TS146  Reference not mentioned in text.
TS147  Reference not mentioned in text.
TS148  Please provide article number or page range.
TS149  Reference not mentioned in text.
TS150  Please provide article number or page range.
TS151  Reference not mentioned in text.
TS152  Reference not mentioned in text.
TS153  Reference not mentioned in text.
TS154  Please provide article number or page range.
TS155  Reference not mentioned in text.
TS156  Reference not mentioned in text.
TS157  Reference not mentioned in text.
TS158  Reference not mentioned in text.
TS159  Reference not mentioned in text.
TS160  Reference not mentioned in text.
TS161  Please provide volume with article number or page range.
TS162  Reference not mentioned in text.
TS163  Please provide article number or page range.
TS164  Reference not mentioned in text.
TS165  Reference not mentioned in text.
TS166  Reference not mentioned in text.
TS167  Please provide article number or page range.
TS168  Reference not mentioned in text.
TS169  Reference not mentioned in text.
TS170  Reference not mentioned in text.
TS171  Reference not mentioned in text.
TS172  Reference not mentioned in text.
TS173  Please provide article number or page range.
TS174  Reference not mentioned in text.
TS175  Reference not mentioned in text.
TS176  Please provide article number or page range.
TS177  Reference not mentioned in text.
TS178  Reference not mentioned in text.
TS179  Reference not mentioned in text.
TS180  Reference not mentioned in text.
TS181  Reference not mentioned in text.
TS182  Reference not mentioned in text.
TS183  Reference not mentioned in text.
TS184  Please provide article number or page range.
TS185  Reference not mentioned in text.
TS186  Reference not mentioned in text.
TS187  Reference not mentioned in text.
TS188  Reference not mentioned in text.
TS189  Reference not mentioned in text.
TS190  Please provide article number or page range.
TS191  Reference not mentioned in text.
TS192  Reference not mentioned in text.
TS193  Reference not mentioned in text.
TS194  Reference not mentioned in text.
TS195  Reference not mentioned in text.
TS196  Please provide article number or page range.
TS197  Reference not mentioned in text.
TS198  Reference not mentioned in text.
TS199  Reference not mentioned in text.
TS200  Reference not mentioned in text.

TS201  Reference not mentioned in text.
TS202  Please provide article number or page range.
TS203  Reference not mentioned in text.
TS204  Reference not mentioned in text.
TS205  Reference not mentioned in text.
TS206  Reference not mentioned in text.
TS207  Reference not mentioned in text.
TS208  Reference not mentioned in text.
TS209  Reference not mentioned in text.
TS210  Please provide article number or page range.
TS211  Reference not mentioned in text.
TS212  Reference not mentioned in text.
TS213  Reference not mentioned in text.
TS214  Reference not mentioned in text.
TS215  Reference not mentioned in text.
TS216  Please provide article number or page range.
TS217  Reference not mentioned in text.
TS218  Reference not mentioned in text.
TS219  Reference not mentioned in text.
TS220  Reference not mentioned in text.
TS221  Reference not mentioned in text.
TS222  Reference not mentioned in text.
TS223  Reference not mentioned in text.
TS224  Reference not mentioned in text.
TS225  Reference not mentioned in text.
TS226  Reference not mentioned in text.
TS227  Reference not mentioned in text.
TS228  Please provide article number or page range.
TS229  Reference not mentioned in text.
TS230  Reference not mentioned in text.
TS231  Please provide article number or page range.
TS232  Reference not mentioned in text.
TS233  Reference not mentioned in text.
TS234  Reference not mentioned in text.
TS235  Reference not mentioned in text.
TS236  Reference not mentioned in text.
TS237  Reference not mentioned in text.
TS238  Reference not mentioned in text.
TS239  Reference not mentioned in text.
TS240  Reference not mentioned in text.
TS241  Reference not mentioned in text.
TS242  Reference not mentioned in text.
TS243  Please provide article number or page range.
TS244  Reference not mentioned in text.
TS245  Reference not mentioned in text.
TS246  Reference not mentioned in text.
TS247  Reference not mentioned in text.
TS248  Reference not mentioned in text.
TS249  Reference not mentioned in text.
TS250  Please provide article number or page range.
TS251  Reference not mentioned in text.
TS252  Please provide article number or page range.
TS253  Reference not mentioned in text.
TS254  Reference not mentioned in text.
TS255  Reference not mentioned in text.

TS256 Reference not mentioned in text.
TS257 Reference not mentioned in text.
TS258 Reference not mentioned in text.
TS259 Please provide article number or page range.
TS260 Reference not mentioned in text.
TS261 Reference not mentioned in text.
TS262 Reference not mentioned in text.
TS263 Please provide article number or page range.
TS264 Reference not mentioned in text.
TS265 Reference not mentioned in text.
TS266 Reference not mentioned in text.
TS267 Please provide article number or page range.
TS268 Reference not mentioned in text.
TS269 Reference not mentioned in text.
TS270 Reference not mentioned in text.
TS271 Reference not mentioned in text.
TS272 Reference not mentioned in text.
TS273 Reference not mentioned in text.
TS274 Reference not mentioned in text.
TS275 Reference not mentioned in text.
TS276 Reference not mentioned in text.
TS277 Reference not mentioned in text.
TS278 Reference not mentioned in text.
TS279 Please provide article number or page range.
TS280 Reference not mentioned in text.
TS281 Reference not mentioned in text.
TS282 Reference not mentioned in text.
TS283 Please provide article number or page range.
TS284 Reference not mentioned in text.
TS285 Please provide article number or page range.
TS286 Reference not mentioned in text.
TS287 Reference not mentioned in text.
TS288 Please provide article number or page range.
TS289 Reference not mentioned in text.