# Peer review of "Integrating palaeoclimate time series with rich metadata for uncertainty modelling: strategy and documentation of the PALMOD 130k marine palaeoclimate data synthesis"

_Earth System Science Data, 2019_

## Referee Comment (RC1) · Anonymous Referee #1 · 29 Jan 2020

The authors present a compilation of paleoclimate data from marine sediment cores covering the past 130 kyr. They give a clear account of their data acquisition strategy, which focussed on cores with d18O measured on benthic foraminifera and radiocarbon dates so that a robust common chronology could be constructed for the entire compilation. Where other paleoclimate data were available for the same core, these data were included in the compilation. They pay close attention to including meta data required to analyse the data further.

Parameter and metadata names have been harmonised and the original naming is

preserved so that these can be traced in the original publications. The data are all well referenced with DOIs and citations. New depth-age models have been constructed for all sites using BACON and the published chronologies preserved for reference.

In all this represents a very well researched and harmonised dataset with rich and useful metadata that does not exist elsewhere. The data are supplied in a variety of formats, as R data objects, NetCDF, and in the LiPD format which is itself a set of zipped plain text (csv) files containing the data in a highly structured JSON format (http://wiki.linked.earth/Linked_Paleo_Data).

However, the structure of the data within the R objects makes it very difficult to search for and extract subsets of data. For example, all the records of variable "planktonic.d18O", or all the records in a certain geographical region. In the NetCDF and LiPD formats the data are also structured in a similar way, although there may be tools available to help work with the LiPD data.

Use of this data compilation would be greatly enhanced if the data were re-structured into a set of "partially normalized tables" in a "star schema" so that queries can be made in an SQL-like way by joining tables and using select and filter type statements. See Brian McGill's 3rd Commandment here (https://dynamicecology.wordpress.com/2016/08/22/ten-commandments-for-good-data-management/). A key table in this format would for example be the "ParameterListWithRefs.csv" table linked to in the Data availability statement of this manuscript but not found within the data objects.

No specific database software needs to be used; these could be plain text files that could be read in by many data analysis software. This is not a "big" dataset so the structure does not need to optimise storage or retrieval efficiency.

I'm not suggesting that reformatting the data in this way would be trivial for the authors, but the data in their current format are well structured and so it should be possible to write code to do it – and this should be much easier for the authors than for someone

coming to it fresh.

Minor comment: l. 486 - In the section "recommendations for data archiving" "Include metadata" I would also recommend including information about the size of the sample on which the parameter was measured, e.g. number of foraminifera, mass of sample, total peak area for Alkenones. As this can be very useful when assessing the uncertainty of the value.

Text errors:

l. 147 "were" -> "where"

l. 161 "more of data"

---

## Referee Comment (RC2) · Blanca Ausin (Referee) · 28 Feb 2020

The authors present a database consisting on paleoclimate records from several climate-sensitive parameters from 143 marine sites spanning the last 130 kyr. New chronologies have been built for each site for consistency while corresponding original chronologies are also provided. One advantage over other data compilations is the careful treatment of chronological data. The database contains rich metadata needed to perform a robust assessment of the paleoclimate signals and is available in three different formats facilitating its use in free software. The manuscript is clearly written and

has a straightforward explanation of the data search and treatment strategy, database structure, data storage and future plans.

The authors provide good rationale on the need of a database of time series data for modelling purposes and the importance of data formatting and standardization and their effort to build the presented database should be welcomed by the paleoclimate and climate modeler community. There are a couple comments I raise below I think the authors should take care of:

Lines 257-265: The use of variable reservoir ages to build age-depth models is a hot topic that is currently under debate. So far, no clear consensus exists on whether the general use of this or the "static reservoir" approach is more robust, advantageous, or beneficial than the other and the choice is made by every author based on different reasons. A discussion on why the authors decide to apply this approach should be included.

Also, I find their specific approach strongly relies on Butzin et al. (2017) model and is limited by the spatial coverage of the latter. How did the authors decide on which reservoir ages to use for sediment cores located out of the original data coverage of Butzin et al. (2017)? What is the uncertainty associated with the reservoir ages taken from the extrapolated regions in Butzin et al. (2017) and do the authors account for it? More specifically, do the authors account for additional uncertainty of reservoir ages for cores retrieved from upwelling regions and semi-isolated basins (if any is included in the database) where the effect of regional processes are not considered by the coarse-resolution of most general circulation models?

SST is a vital component of a vast majority of modelling efforts. I highly encourage the authors to re-calibrate SST estimates from alkenone ratios with the latest global calibration by Tierney and Tingley (2018). This does not differ much from previous calibrations for estimates below 24°, but over that value it corrects for the slope attenuation in the Uk'37-SST relationship providing temperature estimates several degrees different from previous estimates. This calibration also provides error estimates and specific calibrations for regions where seasonality has a strong influence of alkenones-derived SST.

Minor comments: Line 28: Why there are more benthic $\delta$18O time series than sites?

Line 96: What do you mean by homogenous? Please add continuous chronology.

Lines 122 and 123: the term "marine sediment sequences" might be more appropriate than "marine sediment archives".

Line 132: Please state somewhere in this paragraph why the need for combining radiocarbon and benthic $\delta$18O for the chronological control (e.g., radiocarbon limit).

Line 151-156: why so self-critical? Chronostratigraphies and age-depth models are the backbone of paleoclimate data. I find the choice of stating a robust and consistent chronological control as the conditioning criteria to build the stratigraphic framework an advantage rather than a pitfall, even if the database is not as comprehensive as it could be by using more flexible criteria for the time control.

Line 161: rephrase "by including more data".

Line 181: rephrase "we followed the reduction approach".

Line 218: Please mention tables in sequential order.

Line 233: in order to

Line 310: those cases when it is not possible to indicate the calibration are because this is not reported in the original paper?

Line 312: Species and number of individuals are reported as this is important information to assess foraminifera-derived proxy data. The same is true for foraminifera size, which indeed is reported in the database. Please add here the effect of the size of foraminiferal tests on derived measurements.

Line 321: The Uk'37 ratio is not based on alkenones with different chain lengths (both have 37 carbons) but with a different number of unsaturations (di- and tri-unsaturated). Please correct.

Line 323: Please replace: not constant by "seasonal".

Line 326: C37 alkenone concentration is included in the database but its significance as productivity proxy is not included in the discussion. Is there any reason for this?

Line 364: Replace Table by table.

Line 383: Replace table by Table.

Line 428: Does this mean no more 14C-based chronologies? Why?

Line 430: Is there any plan for authors submitting their data for inclusion in the database?

Line 470: The observation time series?

Line 487: Based on planktonic foraminifera Mg/Ca. . . Please rephrase.

Line 513: Replace Pangaea by PANGAEA

Line 518: . . . data consists data not part of publication. Rephrase.

Table 4: it might be convenient to replace "detritus" by "debris" for searching purposes as the latter is more common in the paleoclimate field.

Tierney, J.E., Tingley, M.P., 2018. BAYSPLINE: A New Calibration for the Alkenone Paleothermometer. Paleoceanography and Paleoclimatology 33, 281-301.

---

## Author Response (AR1)

Dear David,

Please find a revised version of our manuscript: "Integrating palaeoclimate time series with rich metadata for uncertainty modelling: strategy and documentation of the PALMOD 130k marine palaeoclimate data synthesis". We would like the thank the reviewers for their helpful feedback on the previous version. We have addressed all their comments and a detailed response, as well as an updated version of the manuscript with tracked changes, is provided below. We feel that the manuscript is now in a better shape and hope that it meets the criteria for publication in *Earth System Science Data*.

In addition to changes to the text we have also improved the standardisation and corrected some minor mistakes (e.g. typos) in the metadata. We have not added any sites, nor added data, so the figures in the manuscript are still up to date. The updated data product (version 1.0.1; https://seafile.zfn.uni-bremen.de/d/c146404ef5634c2497c2/) has been submitted for long-term archiving at PANGAEA. However, due to the current corona crisis, PANGAEA is at the moment receiving more submission whilst at the same time dealing with personnel shortages, which means that the processing time can be up to several weeks. We hope that we can provide a new download link to the data soon and any updates to the data product will also be clearly highlighted at the PANGAEA page where version 1.0.0 is located (https://doi.pangaea.de/10.1594/PANGAEA.908831). As said above, this update does not in any way affect the data themselves and the data descriptor is therefore not affected. We therefore hope that the revised manuscript can be considered for publication at this stage.

With kind regards,

Lukas Jonkers
(on behalf of all authors)

We would like to thank the reviewer for their constructive comments. We copied the comments below and provide our proposed changes in red. We hope that these changes address the concerns and that a revised manuscript will meet the criteria for publication in Earth System Science Data.

Lukas Jonkers
(On behalf of all authors)

The authors present a compilation of paleoclimate data from marine sediment cores covering the past 130 kyr. They give a clear account of their data acquisition strategy, which focussed on cores with d18O measured on benthic foraminifera and radiocarbon dates so that a robust common chronology could be constructed for the entire compilation. Where other paleoclimate data were available for the same core, these data were included in the compilation. They pay close attention to including meta data required to analyse the data further.

Parameter and metadata names have been harmonised and the original naming is preserved so that these can be traced in the original publications. The data are all well referenced with DOIs and citations. New depth-age models have been constructed for all sites using BACON and the published chronologies preserved for reference.

In all this represents a very well researched and harmonised dataset with rich and useful metadata that does not exist elsewhere. The data are supplied in a variety of formats, as R data objects, NetCDF, and in the LiPD format which is itself a set of zipped plain text (csv) files containing the data in a highly structured JSON format (http://wiki.linked.earth/Linked_Paleo_Data).
We appreciate the reviewer's feedback on the value of the data product.

However, the structure of the data within the R objects makes it very difficult to search for and extract subsets of data. For example, all the records of variable "planktonic.d18O", or all the records in a certain geographical region. In the NetCDF and LiPD formats the data are also structured in a similar way, although there may be tools available to help work with the LiPD data.

Use of this data compilation would be greatly enhanced if the data were re- structured into a set of "partially normalized tables" in a "star schema" so that queries can be made in an SQL-like way by joining tables and using select and filter type statements. See Brian McGill's 3rd Commandment here (https://dynamicecology.wordpress.com/2016/08/22/ten-commandments- for-good-data-management/). A key table in this format would for example be the "ParameterListWithRefs.csv" table linked to in the Data availability statement of this manuscript but not found within the data objects.

No specific database software needs to be used; these could be plain text files that could be read in by many data analysis software. This is not a "big" dataset so the structure does not need to optimise storage or retrieval efficiency.

I'm not suggesting that reformatting the data in this way would be trivial for the authors, but the data in their current format are well structured and so it should be possible to write code to do it – and this should be much easier for the authors than for someone coming to it fresh.

We thank the reviewer for their feedback on the format of the database. Because of its structure, linking multiple records to common site information, complex metadata and chronological information, the data we provide are indeed not as easy to query as a simple text file would be. Yet, the referee is right to point out that the ability to extract user-defined sets of records, such as by regions or by proxy type, is a key functionality. We have therefore provided example scripts to query the RDS files and made them available on GitHub (https://github.com/lukasjonkers/PALMODutils). We would like to highlight that similar tools are already available to query LiPD files (https://github.com/nickmckay/LiPD-utilities) and that the PaleoDataView software is specifically designed to interact with the netcdf files (https://www.marum.de/en/Stefan-Mulitza/PaleoDataView.html). Considering that for both the LiPD and netCDF formats querying tools are available, and that we provide similar options for R, we prefer to keep the format of the database as it is now, without shoehorning its highly interlinked structure into another format. This is because the structure of the database also allows for better quality control and updating of the (meta)data and age-depth models. The individual files also make unintentional mixing of information less likely.

The example scripts that we will provide will allow the user to build their own version of a table like 'ParameterListWithRefs.csv' and to query the database by:

- Parameter
- Parameter detail
- Sensor species
- Minimum age
- Maximum age
- Resolution
- Number of tie points

The example code returns the indices of the sites that meet the criteria, which the user can use to extract the desired time series data, metadata or chronology data. This is intentional as custom scripts are required to tailor the data extraction to each analysis. We have updated section 6 and incorporated the information on how to interact with the database.

Minor comment: l. 486 - In the section "recommendations for data archiving" "Include metadata" I would also recommend including information about the size of the sample on which the parameter was measured, e.g. number of foraminifera, mass of sample, total peak area for Alkenones. As this can be very useful when assessing the uncertainty of the value.

We agree and will include a statement encouraging researchers to include those data in the section with recommendations. Note that this also follows the recommendation of PaCTS (Khider et al., 2019).

Text errors:
l. 147 "were" -> "where"
l. 161 "more of data"
These will be addressed.

References:

Khider, D., Emile-Geay, J., McKay, N. P., Gil, Y., Garijo, D., Ratnakar, V., Alonso-Garcia, M., Bertrand, S., Bothe, O., Brewer, P., Bunn, A., Chevalier, M., Comas-Bru, L., Csank, A., Dassié, E., DeLong, K., Felis, T., Francus, P., Frappier, A., Gray, W., Goring, S., Jonkers, L., Kahle, M., Kaufman, D., Kehrwald, N. M., Martrat, B., McGregor, H., Richey, J., Schmittner, A., Scroxton, N., Sutherland, E., Thirumalai, K., Allen, K., Arnaud, F., Axford, Y., Barrows, T. T., Bazin, L., Pilaar Birch, S. E., Bradley, E., Bregy, J., Capron, E., Cartapanis, O., Chiang, H. W., Cobb, K., Debret, M., Dommain, R., Du, J., Dyez, K., Emerick, S., Erb, M. P., Falster, G., Finsinger, W., Fortier, D., Gauthier, N., George, S., Grimm, E., Hertzberg, J., Hibbert, F., Hillman, A., Hobbs, W., Huber, M., Hughes, A. L. C., Jaccard, S., Ruan, J., Kienast, M., Konecky, B., Le Roux, G., Lyubchich, V., Novello, V. F., Olaka, L., Partin, J. W., Pearce, C., Phipps, S. J., Pignol, C., Piotrowska, N., Poli, M. S., Prokopenko, A., Schwanck, F., Stepanek, C., Swann, G. E. A., Telford, R., Thomas, E., Thomas, Z., Truebe, S., von Gunten, L., Waite, A., Weitzel, N., Wilhelm, B., Williams, J., Williams, J. J., Winstrup, M., Zhao, N., and Zhou, Y.: PaCTS 1.0: A Crowdsourced Reporting Standard for Paleoclimate Data, Paleoceanography and Paleoclimatology, 10.1029/2019pa003632, 2019.

We would like to thank Blanca Ausin for her constructive comments. We copied the comments below and provide our proposed changes in red. We hope that these changes address the concerns and that a revised manuscript will meet the criteria for publication in Earth System Science Data.

Lukas Jonkers
(On behalf of all authors)

The authors present a database consisting on paleoclimate records from several climate-sensitive parameters from 143 marine sites spanning the last 130 kyr. New chronologies have been built for each site for consistency while corresponding original chronologies are also provided. One advantage over other data compilations is the careful treatment of chronological data. The database contains rich metadata needed to perform a robust assessment of the paleoclimate signals and is available in three different formats facilitating its use in free software. The manuscript is clearly written and has a straightforward explanation of the data search and treatment strategy, database structure, data storage and future plans.

The authors provide good rationale on the need of a database of time series data for modelling purposes and the importance of data formatting and standardization and their effort to build the presented database should be welcomed by the paleoclimate and climate modeler community. There are a couple comments I raise below I think the authors should take care of:

Lines 257-265: The use of variable reservoir ages to build age-depth models is a hot topic that is currently under debate. So far, no clear consensus exists on whether the general use of this or the "static reservoir" approach is more robust, advantageous, or beneficial than the other and the choice is made by every author based on different reasons. A discussion on why the authors decide to apply this approach should be included.

The reviewer rightly points out the non-trivial issue of reservoir corrections in age-depth modelling. It has been established beyond doubt that reservoir ages vary in space and time, so the assumption of a static reservoir age is not warranted. Its attractiveness is only in its simplicity as it allows for straightforward age-depth modelling, especially given the uncertainty of how reservoir ages varied in the past. Therefore, we feel that our approach is more realistic as it attempts to incorporate the variability of reservoir ages. Because it is based on a model approach, it has the additional advantage that it does so in a globally physically constrained way. Our original wording also reflects this: "To ensure a common chronological framework for all time series in the synthesis, radiocarbon ages were re-calibrated using the IntCal13 curve (Reimer et al., 2013). We used reservoir age estimates (including uncertainty) based on a comprehensive ocean general circulation model (Butzin et al., 2017) to account for physically plausible spatial and temporal variability in the reservoir ages." That said, it is of course not given that the model by Butzin et al. (2017) is the ultimate answer to the question of how reservoir ages varied in the past. This work will be superseded by new studies and indeed in many situations, like for comparison with published work, the user may wish to see age-depth models calibrated to constant reservoir ages of different kinds. We have anticipated this development, which is why all information on age-depth model development is kept in the database and can be used to derive alternative models (see e.g. section 2.3). We felt it important to provide uniformly derived age estimates for all records and believe the approach we use is not inferior to any existing alternative. What we would like to avoid is to overinflate the dataset by providing multiple age-depth models for each site. In the revised manuscript we will better explain our choice to use variable reservoir ages, but as we provide the necessary data for the user to amend the age-depth models we think that a discussion about which approach is more robust is moot/does not add to the manuscript. Given the comment below, we have however also improved the description of how we obtained the reservoir ages and assessed their uncertainty (section 2.3).

Also, I find their specific approach strongly relies on Butzin et al. (2017) model and is limited by the spatial coverage of the latter. How did the authors decide on which reservoir ages to use for sediment cores located out of the original data coverage of Butzin et al. (2017)? What is the uncertainty associated with the reservoir ages taken from the extrapolated regions in Butzin et al. (2017) and do the authors account for it? More specifically, do the authors account for additional uncertainty of reservoir ages for cores retrieved from upwelling regions and semi-isolated basins (if any is included in the database) where the effect of regional processes are not considered by the coarse- resolution of most general circulation models?

The reviewer is correct that our approach to take a variable reservoir age into account is limited by the resolution (3.5°) of the data set from Butzin et al. (2017). In most current studies, marine radiocarbon age models are based on a calibration with the Marine13 calibration curve, which is already corrected by a global modelled reservoir age and can be further corrected with a local dR that is usually assumed to be constant. However, although the Butzin et al. (2017) data set can only be a first-order approximation, we see several advantages in its application: First the dataset allows us to take some reasonable regional variation of the reservoir age into account. For example, the high glacial reservoir ages in the North Atlantic (i.e. Sarnthein et al. 2015, Radiocarbon, 57(1), 129–151, doi:10.2458/azu_rc.57.17916) are well reproduced by Butzin et al. (2017). Second, Butzin et al. (2017) consider the effect of pCO2 changes on the global reservoir age of the surface ocean that may lead to a glacial global reservoir age of 600-700 years compared to a global reservoir age of 405 years taken into account for most of Marine13. To derive the reservoir age/reservoir age error for a measured radiocarbon age we (i) extract all modelled radiocarbon ages from the nearest gridbox in the modelled data set, (ii) find all modelled radiocarbon ages that are possible within the error of the measured radiocarbon age and then (iii) take the mean and the standard deviation of all corresponding reservoir ages as correction for the measured radiocarbon age. Due to a lack of information, we cannot account for local uncertainties in the reservoir ages, i.e. due to upwelling. To clarify, we have added further information, how reservoir ages have been derived (section 2.3). The issue of reservoir ages in semi-enclosed basins such as the Mediterranean and Red Seas, was already described in the original manuscript in section 2.3.

SST is a vital component of a vast majority of modelling efforts. I highly encourage the authors to re-calibrate SST estimates from alkenone ratios with the latest global calibration by Tierney and Tingley (2018). This does not differ much from previous calibrations for estimates below 24◦, but over that value it corrects for the slope attenuation in the Uk'37-SST relationship providing temperature estimates several degrees different from previous estimates. This calibration also provides error estimates and specific calibrations for regions where seasonality has a strong influence of alkenones-derived SST.
We agree with the reviewer that calibration of temperature proxies is an important issue. Precisely for this reason we have included the raw data in the database whenever this was possible to allow the user of the database to recalibrate any temperature proxy using the calibration they want/deem most appropriate. At this stage we refrain from recalibrating UK37 temperature estimates because we plan updates of the database that focus more on temperature. In these updates we will ensure globally consistent calibration of all temperature proxies. In addition, we feel that changing the original calibration would require more consideration than just changing the calibration equation, as the calibration choice of the authors of each individual dataset are likely to have clear reasons for using each specific calibration. We therefore think that recalibrating all temperature proxies goes beyond the scope of the present manuscript. Recalibrating only the UK37 temperatures seems in our opinion arbitrary given that progress has also been made on Mg/Ca temperatures, transfer functions etc. We stress however, that any user of the database can with relatively little effort change the calibration that was used by the authors of the data, as long as raw data were made available.

Minor comments:
Line 28: Why there are more benthic $\delta18O$ time series than sites?
This is a good question. It is because some sites contain data from more than a single species, or multiple time-series from the same species that could not be merged, for instance because of differences in size fraction of the shells used for the measurements.
Line 96: What do you mean by homogenous? Please add continuous chronology.
Clarified. The new sentence reads: "We focus on the ocean as it is a large reservoir of heat and $CO_2$ and allows global coverage with consistent chronological control." L: 96.
Lines 122 and 123: the term "marine sediment sequences" might be more appropriate than "marine sediment archives".
We prefer to keep the original wording.
Line 132: Please state somewhere in this paragraph why the need for combining radiocarbon and benthic $\delta18O$ for the chronological control (e.g., radiocarbon limit).
Good point. We inserted the following sentence: "This approach of blending absolute and relative age controls is required to provide age-depth models for sediment cores that extend beyond the radiocarbon dating range (~40,000 years)." L:10-142.

Line 151-156: why so self-critical? Chronostratigraphies and age-depth models are the backbone of paleoclimate data. I find the choice of stating a robust and consistent chronological control as the conditioning criteria to build the stratigraphic framework an advantage rather than a pitfall, even if the database is not as comprehensive as it could be by using more flexible criteria for the time control.

We agree with the reviewer and are happy with their support for our approach. However, we would like to stick with the current wording because we envision updates of the database that will include different age modelling strategies.

Line 161: rephrase "by including more data".

Done.

Line 181: rephrase "we followed the reduction approach".

Done.

Line 218: Please mention tables in sequential order.

Done.

Line 233: in order to

Done.

Line 310: those cases when it is not possible to indicate the calibration are because this is not reported in the original paper?

Yes, we have made this clearer.

Line 312: Species and number of individuals are reported as this is important information to assess foraminifera-derived proxy data. The same is true for foraminifera size, which indeed is reported in the database. Please add here the effect of the size of foraminiferal tests on derived measurements.

Done.

Line 321: The Uk'37 ratio is not based on alkenones with different chain lengths (both have 37 carbons) but with a different number of unsaturations (di- and tri-unsaturated). Please correct.

Silly mistake. Thank you for pointing this out. We have corrected this.

Line 323: Please replace: not constant by "seasonal".

We prefer our original wording because seasonal suggests that no production occurs outside a specific season, whereas all we want to say is that the abundance of foraminifera is variable over the course of a year - such variability may follow a different pattern than strictly seasonal.

Line 326: C37 alkenone concentration is included in the database but its significance as productivity proxy is not included in the discussion. Is there any reason for this?

The reason to include concentration data was that these data may help to assess the reliability of the temperature estimates, which is now also mentioned in L543-546. Alkenone concentration data can only be assumed to reflect productivity where the overall productivity scales linearly with coccolithophore production.

Line 364: Replace Table by table.

Done..

Line 383: Replace table by Table.

Done.

Line 428: Does this mean no more 14C-based chronologies? Why?

Yes, this is correct. To make clear why we do this we changed the sentence to "To increase the spatio-temporal coverage over the entire 130,000-year time frame of the database, updates of this data product will first aim for quantitative growth of the database by adding more time series with chronological control based on benthic foraminifera $\delta^{18}O$ and absolute age control points other than $^{14}C$."

Line 430: Is there any plan for authors submitting their data for inclusion in the database?

The reviewer raises an important point. Our database is not intended as a replacement of data repositories and we only include data that is publicly available and citable to encourage good data stewardship. This is why we have designed the data synthesis workflow in such a way that for updates of the database data files from PANGAEA and NCDC can be ingested in a straightforward manner. Such updates would benefit hugely from the authors depositing their data in a way that follows the recommendations as spelled out in our manuscript. This applies in particular to the inclusion of metadata and chronological information, which is the main point of our section on data archiving. In this regard we would also like to highlight that NCDC now accepts data in LiPD format, which greatly simplifies the inclusions of metadata and standardisation.

Line 470: The observation time series?

We deleted "observation".

Line 487: Based on planktonic foraminifera Mg/Ca. . . Please rephrase.

We deleted "Mg/Ca".

Line 513: Replace Pangaea by PANGAEA

Done.

Line 518: . . . data consists data not part of publication. Rephrase.

Rephrased.

Table 4: it might be convenient to replace "detritus" by "debris" for searching purposes as the latter is more common in the paleoclimate field.

Ice-rafted detritus results in more hits in Google Scholar, so we prefer to keep this naming. We deem the chance of confusion minimal.

[revised manuscript text omitted]